# Bootstrap Your Conversions:
# Thompson Sampling for Partially Observable Delayed Rewards

**Marco Gigli**[1]                         **Fabio Stella**[1]

[1]Department of Informatics, Systems and Communication, University of Milano-Bicocca, 20126 Milan, Italy

## Abstract

This paper presents a novel approach to address contextual bandit problems with partially observable, delayed feedback by introducing an approximate Thompson sampling technique. This is a common setting, with applications ranging from online marketing to vaccine trials. Leveraging Bootstrapped Thompson sampling (BTS), we obtain an approximate posterior distribution over delay distributions and conversion probabilities, thereby extending an Expectation-Maximisation (EM) model to the Bayesian domain. Unlike prior methodologies, our approach does not overlook uncertainty on delays. Within the EM framework, we employ the Kaplan-Meier estimator to place no restriction on delay distributions. Through extensive benchmarking against state-of-the-art techniques, our approach demonstrates superior performance across the majority of tested environments, with comparable performance in the remaining cases. Furthermore, our method offers practical implementation using off-the-shelf libraries, facilitating broader adoption. Our technique lays a foundation for extending to other bandit settings, such as non-contextual bandits or action-dependent delay distributions, promising wider applicability and versatility in real-world applications.

## 1 INTRODUCTION

Stochastic Multi-armed Bandits (MABs) constitute a highly effective framework for sequential decision making in the presence of uncertainty. The stochastic MAB problem can be seen as a special case of Reinforcement Learning, in which the actions of the agent do not modify the environment in which it acts [Sutton and Barto, 2018].

The simplifying assumption that the actions available to the agent, and consequent rewards, are not affected by past decisions, has proven very effective in a host of real-world applications; among many others, landmark application areas include *clinical trials* [Wu and Wager, 2022a] and, in an industry setting, experimentation for the optimisation of *website content* [Wang et al., 2022] and *digital advertising* campaigns [Vernade et al., 2020a].

In all these settings there is the need to balance *exploration* and *exploitation*: the *agent* in charge of decisions must continuously decide between dedicating resources to gathering more data (exploration) and sticking to the choice that has proven the best so far (exploitation). Collecting more data allows making sharper decisions, but it comes at a cost.

In this paper we will focus on *contextual bandits*. As the name suggests, they allow leveraging contextual information to aid in the learning and decision processes. They are particularly appealing in the mentioned settings, where features representing patients (e.g., age, blood pressure) and web surfers (such as demographic features and interests) can be encoded in numerical vectors. For bandit algorithms in general, and contextual bandits in particular, the interested reader is referred to the introduction by Slivkins [2019].

Among the strategies to tackle the exploration-exploitation dilemma in the bandit setting, one stands out owing to its generality, conceptual simplicity and good practical performance: Thompson sampling [Thompson, 1933, Russo et al., 2017]. The agent holds a posterior distribution over the space of parameters governing the data generation process and updates said posterior as new data become available. When facing a choice between different actions, the agent draws a sample from the posterior distribution, and selects the action which is best according to the drawn sample.

An intrinsic facet of the cited application domains is that of *delayed feedback*. In clinical settings, both intended and side effects often do not surface immediately after treatment. In the online marketing setting, both when optimising website content and advertising campaigns, the goal is most often to maximise customer contacts or, if at all possible, sales,

rather than the number of clicks, as the latter is a sub-optimal proxy; while a click usually occurs within minutes since the user is shown content, the delay between first click and contact or sale (two events that can be collectively called *conversions*) can reach several weeks and cannot be ignored.

Among the rich literature on delayed rewards in stochastic bandits, the context of online marketing stands out, in that the feedback is only *partially observable*. As remarked by Chapelle [2014], while positive feedback (purchase decisions) becomes accessible to the agent after some delay, negative feedback is never explicitly observed; in other words, the agent cannot distinguish between users who do not convert and those who have not converted yet, but will do so in the future. Clinical trials of vaccines, too, can be affected by the same problem [Wu and Wager, 2022a]. Also challenging is the fact that delay distributions are often heavy-tailed [Diemert et al., 2017] and poorly modelled by parametric distributions.

## 1.1 CONTRIBUTIONS

In this paper, we introduce a novel approximate Thompson sampling technique to treat contextual bandits with partially observable, delayed feedback. This technique leverages Bootstrapped Thompson sampling (BTS) [Osband and Van Roy, 2015] to build an approximate posterior distribution over delay distributions and probabilities of conversion, thus extending an Expectation-Maximisation (EM) model [Chapelle, 2014] to the Bayesian domain. In doing so, we treat on the same footing uncertainty on delays and on conversions: in previous attempts, the uncertainty on delays is ignored [Wang et al., 2022]. BTS has the advantage of requiring minimal assumptions and being way faster than Markov chain Monte Carlo. To the best of our knowledge, this is the first time BTS is used in conjunction with EM. To treat non-parametric delay distributions, the underlying EM model is used with the Kaplan-Meier estimator (a standard Maximum Likelihood non-parametric estimator for time distributions) for the first time.

The proposed approach is benchmarked against the state of the art [Vernade et al., 2020a] on a host of delay distributions and is found performing significantly better in the vast majority of cases, behaving comparably in the remaining ones. As another advantage, the proposed approach can be readily implemented with off-the-shelf libraries, as shown below. Finally, the technique can be easily generalised to other bandit settings (e.g., non-contextual bandits or action-dependent delay distributions).

## 1.2 RELATED WORK

There is a rich literature on stochastic MABs with delayed feedback: after the foundational works [Chapelle and Li, 2011, Dudík et al., 2011, Joulani et al., 2013, Mandel et al.,

2015], effort has concentrated on specific streams such as Gaussian Process bandits with batch updates, aggregated anonymous feedback, generalised-linear bandits and dealing with intermediate feedback: see respectively the works by Verma et al. [2022], Wang et al. [2021], Howson et al. [2023] and McDonald et al. [2023] for recent pointers to the literature.

In this paper, we focus on linear contextual bandits with partially observable, delayed feedback, i.e., the setting studied by Vernade et al. [2020a]. Their approach presents room for improvement, since it ignores data about the magnitude of the delays: when available, this added information can increase performance. Moreover, their proposed method involves setting beforehand a number of rounds $m$ beyond which the reward is labelled as negative. While conceivably such window parameter may be externally imposed, in all other cases it is not clear how it could be tuned, without knowledge of the distribution of delays. Additionally, besides maximising the number of conversions, practitioners are often interested in the estimate of the probability of conversion itself [Wang et al., 2022], while in their work this estimate is biased by design. Finally, their approach requires carefully deriving Upper Confidence Bounds: it is non-trivial to extend it to other bandit settings (e.g., non-linear contextual). While they also present a sampling approach, it is an *ansatz* based on the same bounds.

The closest approach to the one here proposed is given by Wang et al. [2022], which present a TS-like technique. Starting from the EM model of Chapelle [2014], they re-weight observations in view of the learned distribution of delays. In this way, the problem is mapped to a standard Bernoulli bandit, and TS is carried out through a Beta-Bernoulli conjugate pair. This means, however, that they only consider (via a Beta posterior) uncertainty on the distribution of rewards, ignoring uncertainty on the learned distribution of delays: this could lead to insufficient exploration. Moreover, they consider finite-armed bandits, while we aim at contextual bandits. Furthermore, while their model is derived for a general delay distribution, they then use an exponential distribution in experiments; we aim at handling a generic distribution, since in some practical cases the distribution of delays is far from exponential [Diemert et al., 2017].

The context of the work by Lancewicki et al. [2021] presents some similarities with ours: they consider the case of reward-dependent delays, in which realised delays may depend on the stochastic rewards. Partially observable rewards can be in fact alternatively formulated in terms of a non-factorised probability distribution $p(C, D)$ over conversions $C$ and delays $D$; with a slight abuse of notation, $p(D|C = 0) = \delta(D - \infty)$, the Dirac delta concentrated at positive infinity. However, the authors assume *full observability* instead: if one tries and apply their approach to this rather extreme case of delay-reward dependence, one gets a degenerate model: the empirical mean of observed rewards is identically equal

to one for all available actions.

Both Lancewicki et al. [2021] and Wu and Wager [2022b] stress the importance of being able to handle heavy-tailed distributions. On the other hand, also the work by Wu and Wager [2022b] cannot be readily applied to our setting, as the authors explicitly exclude reward-delay dependence, and pure TS was empirically proven suboptimal in the partially observable setting by Wang et al. [2022].

The feedback structure investigated by Wu and Wager [2022a] in the context of vaccine trials is similar to ours, since only negative feedback is observed (infections). However, they concentrate on being able to handle time-dependent risk: they do so at the expense of being able to model the dependence of risk on the time since exposure (to the vaccine in their case, to the advert in online marketing). In other words, they restrict the space of possible delay distributions: if baseline risk were constant, this would correspond to exponential distributions. Moreover they too, as Lancewicki et al. [2021] and Wu and Wager [2022b], work in the discrete case (bandits with a finite number of arms).

Han and Arndt [2021] deal with delayed conversions substituting missing rewards with *surrogate rewards*, generated as follows. Several models are trained to predict the probability of conversion before fixed time-horizons. A (properly rescaled) logistic regression is then trained on the predictions of these models, to extrapolate to times other than the fixed time-horizons: the predictions of this meta model are the surrogate rewards fed to the agent of the bandit problem. Although non-standard, this procedure effectively estimates the dependence on time of the Cumulative Density Function (CDF) of rewards. However, as was the case for the work by Wang et al. [2022], the Bayesian uncertainty of this estimated CDF is not taken into account when applying TS downstream. Moreover, the use of logistic regression places a strong assumption on the distribution of delays (logistic distribution). In what follows, we will properly treat the estimation of the probability of conversion over time as a Survival Analysis problem.

## 2 SEMI-PARAMETRIC MODEL FOR DELAYED CONVERSIONS

### 2.1 PROBLEM SETUP

In this section we describe the data generation mechanism for a contextual bandit with partially observable, delayed feedback. We focus on the website optimisation setting for concreteness, but the concepts hereby exposed can be easily mapped to the other settings mentioned in section 1.

Whenever a new user comes to the website, a new round of the optimisation begins: the agent must decide among $K$ page variants, which constitute the available set of actions

for that round. We assume the agent observes a context $x_A \in \mathbb{R}^d$ for every action $A = 1, \ldots, K$, prior to choosing the action. These contexts describe the page variants and, optionally, the user itself.

We assume that, right after the agent shows a page to the user, two latent variables are generated: the *reward*, a Boolean variable $C$, which indicates whether the user does convert or not, irrespective of when (see section 1 for examples of conversions); a latent time variable $D$, the delay between being shown the page and the conversion (undefined or infinite if $C = 0$). Under this setting the goal of the agent is maximising the number of conversions.

After a time $\delta$ has passed since the agent-user interaction, the conversion may or may not have happened already: we thus introduce a Boolean variable $Y$, which indicates if a conversion has been observed by the agent after the elapsed time $\delta$.

We also introduce a time variable $T$ which aims at capturing all the information available on the time between a past decision (showing a web page) and user's feedback, if any:

$$T = \begin{cases} \delta & \text{if } Y = 0 \\ D & \text{if } Y = 1 \end{cases}$$

If feedback was received (the user has converted), the agent records this time ($T = D$); if there is no feedback yet, the agent records the tightest lower bound available (the time between the action and the current round, $T = \delta$).

To fix the notation, let us call $\vartheta$ the parameters that characterise the probability of conversion $p(C|x, \vartheta)$ and $\eta$ the ones characterising the distribution of delays $p(D|x, \eta)$. Our goal is estimating $\eta$ and especially $\vartheta$ given the available data. We are not making here any assumption about the shape of $\eta$: it could be anything from the rate parameter of an exponential distribution to the list of discrete hazards of a non-parametric Kaplan-Meier estimator; we leave it unspecified, as this technique can be used "plug and play" with the estimator of choice, depending on the experimenter's belief (or lack thereof) on the shape of the delay distribution.

### 2.2 EXPECTATION-MAXIMISATION TECHNIQUE

The EM technique [Dempster et al., 1977] is an iterative procedure, which starts from rough estimates $\vartheta_0, \eta_0$ of $\vartheta$ and $\eta$ and yields gradually refined estimates $\vartheta_k, \eta_k$ as the iteration index $k$ grows. We follow loosely the derivation by Chapelle [2014], but maintain the treatment general with respect to delay distributions and dependency on covariates, so that we will be able to extend the technique to include the non-parametric Kaplan-Meier estimator. In what follows, the dependency on $x$, $\vartheta$ and $\eta$ will be omitted for simplicity.

Iteration $k$ is split between an *expectation step* and a *maximisation step*. In the expectation step, we calculate the

expected value of the log-likelihood $l(Y, T, C)$:

$$Q^{(k)} = l(Y, T, C = 1)p^{(k)}(C = 1|Y, T)$$
$$+ l(Y, T, C = 0)p^{(k)}(C = 0|Y, T),$$

where $p^{(k)}(C|Y, T)$ is calculated using the current estimates (at step $k$) of the parameters $\vartheta$ and $\eta$, starting with a guess at step $k = 0$. In the maximisation step, we maximise the sum of $Q^{(k)}$ over all data points. In this section we focus on the expectation step, while section 2.3 is devoted to the maximisation step.

We start from the calculation of $p^{(k)}(C|Y, T)$: we omit the index $k$ to avoid burdening the notation. If we calculate $w(Y, T) = p(C = 1|Y, T)$, then $p(C = 0|Y, T)$ can be readily obtained, since they sum to one. We first note that $w(1, T)$ is trivially equal to one. Applying Bayes' theorem, we then get:

$$w(0, T) = \frac{p(Y = 0|C = 1, T)p(C = 1)}{p(Y = 0|T)}.$$

By the law of total probability, the denominator becomes

$$p(Y = 0|T) = p(Y = 0|C = 1, T)p(C = 1)$$
$$+ p(Y = 0|C = 0, T)p(C = 0).$$

The probability $p(Y = 0|C = 1, T)$ is just the *survival function* $S(T)$ of the distribution of delays (i.e., the complement to 1 of its CDF), while $p(Y = 0|C = 0, T)$ trivially equals one. We finally call $p_1 = p(C = 1)$ and obtain:

$$w(Y, T) = Y + (1 - Y)\frac{p_1 S(T)}{1 - p_1 + p_1 S(T)}. \quad (1)$$

We now turn to calculating the log-likelihood. The likelihood of observations for which $C = 0$ is just $1 - p_1$, the probability of not converting. When $C = 1$ instead, we get the standard survival analysis likelihood [Harrell, 2015], weighted by $p_1$. If we denote by $f(t)$ the probability density of delays at time $t$, taking the logarithm and re-arranging terms we finally get:

$$Q = \underbrace{w(Y, T)\left[(1 - Y)\log S(T) + Y \log f(T)\right]}_{\text{weighted survival log-likelihood}}$$
$$+ \underbrace{w(Y, T)\log p_1 + (1 - w(Y, T))\log(1 - p_1)}_{\text{weighted classifier log-likelihood}}. \quad (2)$$

We have thus proven that the quantity we want to maximise at step $k$ is composed of two terms: one has the shape of a weighted classifier log-likelihood, while the other is the weighted log-likelihood of a Survival Analysis problem. Note that the two log-likelihoods are decoupled: we can maximise the former with respect to $p_1$ (our next estimate of the probability of conversion), and the latter with respect to $S(t)$ (our next estimate of the survival function of delays).

We also note that the two weighted log-likelihoods have a transparent interpretation. In the survival one, users which did not convert are weighted by our current belief that they will convert in the future, while users that did convert enter fully in the estimation of the distribution of delays. Similarly, in the binary cross-entropy, users that did not convert contribute partially to the component with positive label, and partially to the component with zero label.

Finally, we again remark that we made no assumptions regarding the dependency of $p_1$ and $S(t)$ on covariates $x$. As such, depending on the problem at hand, this formulation can handle univariate models (i.e., no covariates are included), the division of the population into groups (with multiple univariate models, one for each group), and regression models, both linear and non-linear.

## 2.3 FITTING THE MODEL

In this subsection we describe in detail how the two log-likelihoods on the right hand side of (2) can be maximised given a dataset. We remark that, while we did not indicate this explicitly to avoid burdening the notation, at step $k + 1$ the weights $w(Y, T)$ in formula (2) are calculated with the previous estimates $\vartheta_k$ and $\eta_k$, and are thus known. The sum of $Q$ over all data points is then maximised to obtain the next estimates $\vartheta_{k+1}$ and $\eta_{k+1}$.

We start from the classifier log-likelihood. From equation (2) we see that we recover a standard classification problem if we treat each observation in the dataset *as if* it were represented by two observations, one with label $y_1 = 1$ and weight $w_1 = w(Y, T)$, and another with label $y_0 = 0$ and weight $w_0 = 1 - w(Y, T)$:

$$w(Y, T)\log p_1 + (1 - w(Y, T))\log(1 - p_1)$$
$$= w_1\left[y_1 \log p_1 + (1 - y_1)\log(1 - p_1)\right]$$
$$+ w_0\left[y_0 \log p_1 + (1 - y_0)\log(1 - p_1)\right].$$

Therefore, by duplicating the observations and assigning them the appropriate weights and labels, we can utilize the supervised learning algorithm of our choice on this equivalent dataset. In fact, we remark that only the *unobserved* observations (i.e., those with $Y = 0$) need to be duplicated, since when $Y = 1$ the weight $(1 - w(Y, T))$ vanishes. We also remark that, if $p_1$ does *not* depend on a context $x$ (e.g., because we are calculating the probability associated with a given arm in a vanilla MAB, as in the work by Wang et al. [2022]), the classifier log-likelihood is simply maximised by the average of weights $w(Y, T)$.

We now turn to the weighted survival log-likelihood of (2). Here we do not need to transform the dataset in any way: once we choose a Survival Analysis model for $S(T)$ we just need to feed it with the dataset as-is, with weights $w(Y, T)$ defined in (1). In simulations (see section 4) we adopted the Kaplan-Meier estimator which, being non-parametric, is

agnostic with respect to the shape of the delay distribution: although being a classic technique, it is still the de-facto workhorse for non-parametric Maximum Likelihood estimation of censored time data, when dealing with univariate models (i.e., no dependency of the distribution of delays on context). The choice fell on a non-parametric estimator since a strength of the proposed method is being distribution-agnostic: as we will see it works well on a wide range of distributions, without the need to know them beforehand. Finally, besides its theoretical underpinnings, the Kaplan-Meier estimator has the non-negligible advantage of being implemented in off-the-shelf Machine Learning libraries, favouring reproducibility and adoption by practitioners.

On the other hand, the model is able to accommodate also context-dependent delay distributions, $S(T) = S(T|x)$, as in the works by Gael Manegueu et al. [2020], Lancewicki et al. [2021], Wu and Wager [2022b], Wang et al. [2022]. To take into account linear dependence of hazard on context, it would suffice to substitute the Kaplan-Meier estimator with the Cox proportional hazards model [Cox, 1972, Harrell, 2015], which is a semi-parametric model: it makes a linear assumption on the effect of the features on the hazard function, but makes no assumption regarding the nature of the baseline hazard function itself, like the Kaplan-Meier estimator.

As a final remark, note that the termination condition of the EM loop is not specified on purpose: it can be set as reaching a pre-determined number of iterations, or it can entail checking whether the change from $\vartheta_k$ to $\vartheta_{k+1}$ is below some threshold.

# 3 BOOTSTRAPPED THOMPSON SAMPLING FOR DELAYED CONVERSIONS

After having outlined, in the previous section, the data generating process and an appropriate Maximum Likelihood Estimator (MLE) for handling partially observable delayed rewards, we will now cover the major contribution of this work, a novel extension to the bandit setting.

## 3.1 BTS REFRESHER

For a self-contained exposition, we report here the main ideas that power BTS in a general setting. BTS [Eckles and Kaptein, 2014] is an effective technique for approximate sampling from the posterior distribution, as required by Thompson sampling. A number of approaches to BTS exist (see for instance the introduction by Russo et al. [2017] and the references therein). With an eye to practitioners, a version was chosen for the present work, which does not require further calculations besides the MLE, and is closest in spirit to the well known Statistical Bootstrap method

[Efron, 1979, Rubin, 1981]: the approach we will follow is an extension to our setting of the one presented by Osband and Van Roy [2015] for the Bernoulli bandit.

Given an estimator and a dataset, the Statistical Bootstrap yields an estimate of the uncertainty on the estimated quantities (in our case, the probability function $p_1(x)$ and the survival function $S(t)$) by resampling the dataset with replacement many times, and applying the estimator on the resampled data. The main idea behind BTS is sampling from the outcoming distribution of estimates, *as if* they represented the posterior distribution. Indeed, one can show [Rubin, 1981] that the bootstrapped distribution approximates the posterior given a non-informative prior.

However, for this to work, the empirical distribution function of the observed data should approximate reasonably well the population distribution: this assumption breaks down when the size of the dataset is very small. In a bandit problem, this is precisely what happens in the first rounds: underestimating the uncertainty in the first few rounds leads to exploring insufficiently, and exploiting sub-optimal actions. This was proven by Osband and Van Roy [2015] for the Bernoulli bandit: blindly applying the Statistical Bootstrap leads to a regret which, on average, grows linearly with the horizon $T$. That work contains, however, also a simple heuristic solution: enrich the history of played arms and rewards with an *artificial history*, generated from a distribution which can be thought of as a "prior" of sorts. The way these artificial data points are generated is problem specific, so we will now go through the experimental setting.

## 3.2 ENVIRONMENT SETUP

Since, as seen in section 1.2, the state-of-the-art competitors in the considered setting are the algorithms OTFLinUCB and OTFLinTS of Vernade et al. [2020a], for a fair comparison we adopted the same data generation mechanism. We describe it here as, in section 3.3, we will specialise BTS to this setting.

The environment is described by a vector $\vartheta \in \mathbb{R}^d$, with $\|\vartheta\|_2 \leq 1$. At every round, the agent receives from the environment a context $x_A \in \mathbb{R}^d$ with $\|x_A\|_2 \leq 1$ for each action $A = 1, \ldots, K$. The scalar product with $\vartheta$ belongs to the unit interval: $x_A \cdot \vartheta \in [0, 1]$. The reward is then sampled from a Bernoulli distribution with mean $x_A \cdot \vartheta$.

In their experiments, Vernade et al. [2020a] choose $d = 5$ and $K = 10$. Moreover, the environment vector $\vartheta$ is fixed at $\vartheta = (1/\sqrt{d}, \ldots, 1/\sqrt{d})$. Finally, the contexts are sampled independently at each round from $[0, 1]^d$ and then normalised.

As for the delays, Vernade et al. [2020a] tested two distributions: a geometric distribution with varying mean, and an empirical distribution fitted with a Gaussian kernel on the

dataset released by Diemert et al. [2017]. Regarding these real data, in the code accompanying their work, Vernade et al. [2020a] conclude that the delay distribution does not significantly depend on the context, and we will make the same assumption here too. We extended this set of distributions to include all the IID distributions considered by Wu and Wager [2022b] to test pure Thompson sampling in the presence of (fully observable) delays (constant, deterministic delays, uniformly distributed delays over some interval, $\alpha$-Pareto distribution and packet-loss distribution) and Student's $t$ distribution.

## 3.3 ALGORITHM DESCRIPTION

In the setting described in section 3.2, the following mechanism for generating the artificial history was used across all experiments. For every round, $n_{\text{prior}}$ data points were generated. Since these prior-like points have the only goal of making the learner aware that the observed data may not represent the whole population, $n_{\text{prior}}$ was kept way smaller than the horizon $T$, which is greater than 1000 rounds in all experiments: we chose $n_{\text{prior}} = 10$ across all experiments.

Given the assumptions on the environment $\vartheta$ and the contexts $x_A$ above, the following artificial history generation process was deemed natural: for each round, and for each $j = 1, \ldots, n_{\text{prior}}$, both a $\vartheta_j$ and a $x_j$ were generated uniformly over $[0, 1]^d$ and then normalised. The reward was then drawn from a Bernoulli distribution with mean $\vartheta_j \cdot x_j$. Finally, delays were sampled uniformly over $[0, D_{\text{max}}]$, for a $D_{\text{max}}$ lower than the horizon. Since in all experiments the BTS algorithm was compared with OTFLinUCB and OTFLinTS, both of which require a time parameter $m$ (see section 1.2), it seemed natural to fix $D_{\text{max}} = m$. It must be stressed, however, that the parameter $m$ for the OTFLinUCB and OTFLinTS is an integral part of the algorithm and, if it is not externally imposed on the algorithm for memory reasons, it should be tuned accurately depending on the expected delay distribution: we will see below that the performance of these two algorithms is heavily dependent on its value. On the other hand, BTS was found to be roughly independent from the value of $D_{\text{max}}$.

We can thus recap the BTS algorithm at round $n$ (the proposed algorithm is described in detail in Algorithm 1):

- The agent receives a dataset of $n$ observations, each of which contains the observable information explained in subsection 2.1;

- The agent draws $n_{\text{prior}}$ artificial data points, according to the procedure just described;

- The artificial and real data points are merged to form a single dataset of size $(n + n_{\text{prior}})$;

- The agent samples the entire dataset once with replacement;

- The model described in section 2 is fit on this sampled dataset, yielding an estimate $\hat{S}(t)$ and $\hat{p}_1(x)$;

- The agent plays the action $A$ such that, among $x_1, \ldots, x_K$, the context $x_A$ maximises the estimated reward probability $(1 - \hat{S}(T, x))\hat{p}_1(x)$.

---

**Algorithm 1** BootstrapLinTS for partially observable delayed feedback

---

**Input:** $n_{\text{prior}}, D_{\text{max}}, T, d, K$.

1: Data $D_0 = ()$
2: **for** $n = 1, \ldots, T$ **do**
3:      Update data $D_n$ with observed conversions
4:      **for** $j = 1, \ldots, n_{\text{prior}}$ **do**
5:          Sample prior $\vartheta_j$ and $x_j$ uniformly over $[0, 1]^d$
6:          Normalise sampled $\vartheta_j$ and $x_j$
7:          Sample prior reward from Bernoulli($\vartheta_j \cdot x_j$)
8:          Sample delays uniformly over $[0, D_{\text{max}}]$
9:      **end for**
10:     Concatenate $n_{\text{prior}}$ times and rewards with $D_n$
11:     Sample with replacement $n + n_{\text{prior}}$ data points
12:     Estimate $\hat{S}(t, x)$ and $\hat{p}_1(x)$ via EM
13:     Observe current contexts $x_A, A = 1, \ldots, K$
14:     **for** $A = 1, \ldots, K$ **do**
15:        Calculate probability $(1 - \hat{S}(T, x_A))\hat{p}_1(x_A)$
16:     **end for**
17:     Select arm $\text{argmax}_i (1 - \hat{S}(T, x_A))\hat{p}_1(x_A)$
18: **end for**

---

We must remark that the model described in section 2 suffers from the identifiability issue explained by Gael Manegueu et al. [2020]. Namely, two problem instances can produce the same data but have strictly different parameters. As an example, consider problem instance $\mathcal{I}_1$ with, at round $t_1$, $S(t_1) = 80\%$ (i.e., 80% of conversions happen after $t_1$) and $p_1 = 90\%$. Consider then problem instance $\mathcal{I}_2$ with $S(t_1) = 10\%$ and $p_1 = 20\%$. At $t_1$, despite having very different parameters, these two instances produce exactly the same data, as the probability of observing a reward before $t_1$ is given by the product between $p_1$ and the CDF of delays $(1 - S(t_1))$.

Hence, the MLE described above could either output $\mathcal{I}_1, \mathcal{I}_2$ or any other instance which is compatible with the observed data. Nevertheless, the *product* of the predicted conversion probability $\hat{p}_1(x)$ and the predicted CDF $(1 - \hat{S}(t))$ is the same for all these instances: the estimated probability of converting before a certain time is well-identified. This means that the next action should be selected on the basis of this product of probabilities. However, since we are dealing with distributions of delays which do not depend on context, maximising the product or just $p_1(x)$ yields the same result. Extra care should be taken if delays depend on the context. If one (as Wang et al. [2022]) is interested in the actual value of the conversion rate (besides its use for the optimisation algorithm), this should be intended as *conversion probability*

*before a given time.*

# 4   SIMULATION RESULTS

In what follows we will go through the results of the simulations.[1] Two settings have been treated separately. In one, the window parameter $m$ (see section 1.2) is externally imposed on the algorithm: if delay exceeds $m$ rounds, the agent never receives feedback; we call this setting *censored*. In the other setting, $m$ is just a specific of the algorithm for OTFLinUCB and OTFLinTS, and the proposed BTS is free to use all past data: we call this setting *uncensored*. We will see that, as expected, censoring damages the performance of BTS and, among censored variants, the lower $m$ is, the higher the regret. On the other hand, the effect of $m$ on the algorithms of Vernade et al. [2020a] is harder to predict. For every setting we will cover, BTS is among the best performing algorithms, while OTFLinTS is among the worst. This is a reminder that it is not just the act of sampling from a distribution, but also the details of how the distribution is built, that make Thompson sampling an effective technique. For this reason and to avoid clutter, OTFLinTS will not be shown in the following, and BTS will be compared to OTFLinUCB alone. In the plots, we will show the average regret suffered by each algorithm over the course of 20 simulations, together with the standard deviation of the mean. We acknowledge that, in some cases, the error bars overlap and this may slightly diminish readability: due to the slow dependency of standard deviation on the number of replications, significantly reducing their size was incompatible with the time constraints; nevertheless, we remain confident that the plots effectively convey our main points.

**Geometric distribution**  The first distribution we will consider is the geometric distribution with varying average delay. In all three cases, we see from figure 1 that BTS, either uncensored or censored with $m = 500$, performs best. On the other hand, due to the heavy censoring, BTS with $m = 100$ incurs higher regret. Nevertheless, when the average delay equals 100 rounds, its regret is lower with respect to both instances of OTFLinUCB for half of the rounds, and is comparable at the horizon $T = 3000$. In the other two cases, it behaves significantly better than the instance of OTFLinUCB that has access to the same amount of information. Oddly enough, OTFLinUCB with $m = 100$ behaves better, when the average delay equals 100 rounds, with respect to OTFLinUCB with $m = 500$, despite having access to less information: this is due to the way $m$ enters the algorithm. We can thus conclude that, if $m$ is not externally imposed, its choice is non trivial for OTFLinUCB. On the other hand, it is straightforward

for BTS: the higher, the better; if at all possible, it is even better not to censor the feedback.

**Fixed delays**  With fixed, deterministic delays, simulations show that, whenever the censoring time is lower than the delay, the regret grows linearly: this is of course expected, as the agent is completely blind to feedback. Among the other algorithm instances, BTS reaches significantly lower regret. The regret plot is included in the Supplementary Material.

**$\alpha$-Pareto distribution**  The $\alpha$-Pareto distribution presents polynomial tails: the smaller the parameter $\alpha$ is, the heavier is the tail. This is reflected in figure 2, where regret is generally higher for $\alpha = 0.2$: learning takes longer. Besides this, again we see that BTS performs best for all the examined values of $\alpha$. For $\alpha = 0.2$ and $\alpha = 0.5$, the heavily censored instance of BTS is slightly worse than the others, as expected. On the other hand, OTFLinUCB with $m = 500$ incurs much higher regret with respect to the other algorithms (even if it suffers a lower degree of censorship with respect to $m = 100$).

**Packet loss**  By "packet loss", we refer to a scenario in which feedback can be either delivered immediately (i.e., with zero delay) or "lost" (i.e., it has infinite delay). In particular, it is lost with probability $p$ (we employ a definition which is opposite to that used by Lancewicki et al. [2021] and Wu and Wager [2022b], as our $p$ is their $1 - p$). Also in this setting, the instances of BTS behave generally better than OTFLinUCB, and in particular the instance of OTFLinUCB with $m = 500$ performs significantly worse. The regret plot is included in the Supplementary Material. We also note that, for high probability $p$ of lost packet, all algorithms are still learning upon reaching the horizon $T = 3000$: at $p = 0.75$ only one in four events produces observable feedback. Likely, in this setting a different data generating mechanism with respect to that of section 2, that better captures this scenario, could be employed with better results.

**Uniform distribution**  The result is very similar to that of deterministic delays. The regret plot is included in the Supplementary Material.

**Student's $t$ distribution**  To test the proposed algorithm in another heavy-tail environment besides the $\alpha$-Pareto distribution we considered Student's $t$ distribution, varying both the number of degrees of freedom and the scale: since we are dealing with positive delays, we take the absolute value after sampling. For a fixed scale, changing the number of degrees of freedom does not change the results significantly, so we fixed the number of degrees of freedom to 1 (thus reducing to the Cauchy distribution). With a scale parameter equal to 1, OTFLinUCB with $m = 500$ performs significantly worse than the other considered algorithmic variants, which

---

[1]The code is available at `https://github.com/MarcoGigli/bootstrap-conversions`.

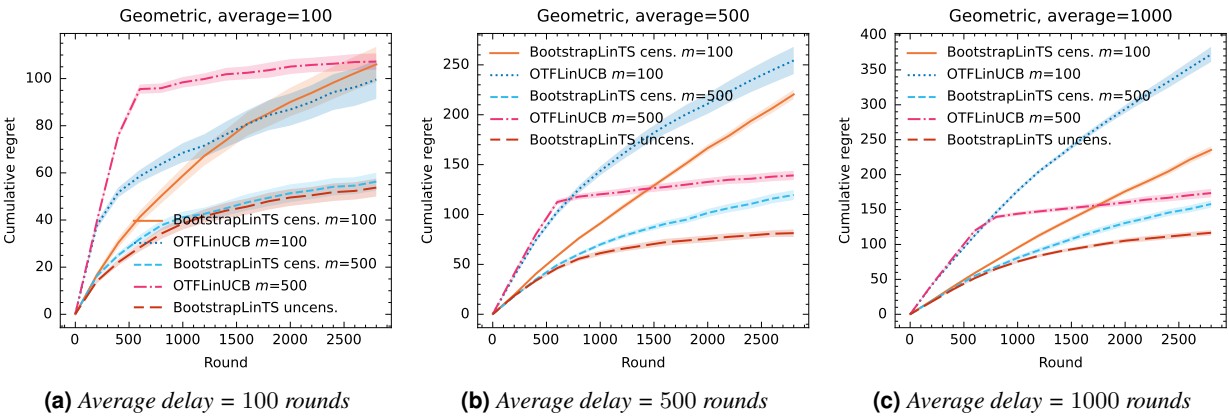

**(a)** *Average delay = 100 rounds*     **(b)** *Average delay = 500 rounds*     **(c)** *Average delay = 1000 rounds*

**Figure 1:** Average cumulative regret suffered by the examined algorithms when delays distribute according to a geometric distribution.

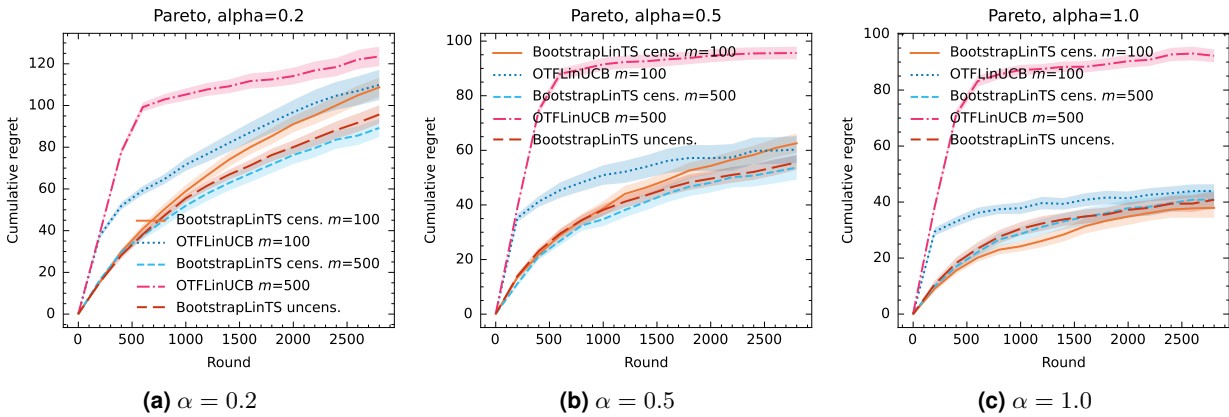

**(a)** $\alpha = 0.2$     **(b)** $\alpha = 0.5$     **(c)** $\alpha = 1.0$

**Figure 2:** Average cumulative regret for delays distributed according to an $\alpha$-Pareto distribution, with varying $\alpha$.

have a comparable performance. Setting the scale to 100, censored BootstrapLinTS with $m = 100$ suffers from the high degree of censorship: it is enough to raise $m$ to 500 to have a performance comparable to the uncensored variant. Setting the scale to 500 yields a similar outcome, but in this case OTFLinUCB with $m = 100$ performs significantly worse than all the other variants. The regret plot is included in the supplementary material.

**Criteo data** As a final setting, we have tested the algorithms on delays distributed according to the Criteo dataset [Diemert et al., 2017]. This dataset contains the recorded delays between click and conversion of digital marketing campaigns. To provide a fair comparison, we used the same model for sampling these delays used by Vernade et al. [2020a], and also the same value of $m$ and of the horizon $T = 10000$. Also in this real-world setting, the results are strikingly in favour of the proposed BTS (figure 3).

**Sensitivity test** In all the experiments described so far, we kept the artificial dataset size $n_{\mathrm{prior}}$ fixed: as explained

in section 3.3, it was chosen to be way smaller (two orders of magnitude) than the horizon $T$. The logic is that, if one and the same parameter choice yields good performance across many environments, one can safely take the chosen configuration as part of the algorithm. On the other hand, it is interesting to see how performance changes varying $n_{\mathrm{prior}}$: for this reason we performed a sensitivity test in one setting (geometric distribution of delays with average 100). At odds with the rest of the experiments, we let $\vartheta$ vary uniformly across experiments: one can verify that with the fixed $\vartheta$ described in section 3.2, BootstrapLinTS is unfairly favoured by a growing $n_{\mathrm{prior}}$. In this setup, the performance for $n_{\mathrm{prior}} = 10$ is very similar to that with a fixed $\vartheta$. Interestingly, performances with $n_{\mathrm{prior}} = 1$ and $n_{\mathrm{prior}} = 10$ are very similar (we show only the former in plots). The performance degrades for $n_{\mathrm{prior}} = 100$: the proposed algorithm incurs lower regret with respect to OTFLinUCB for the whole duration of the experiments, but attests at comparable regret at the horizon $T$. Finally, for $n_{\mathrm{prior}} = 1000$, the proposed algorithm per-

forms worse than OTFLinUCB: this is to be expected, as the size of the artificial random dataset is comparable to that of the real dataset. The plots are included in the supplementary material.

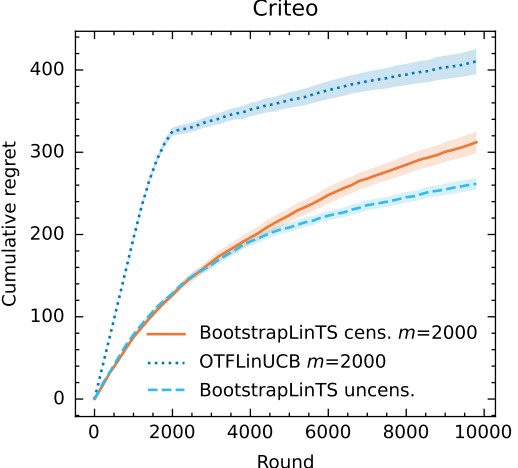

**Figure 3:** Average cumulative regret for delays distributed according to the Criteo dataset [Diemert et al., 2017].

## 5 CONCLUSION

### 5.1 SUMMARY OF THE CONTRIBUTIONS

In the present paper, we have concentrated on sequential optimisation for linear contextual bandits with partially observable delayed rewards. This setting is particularly relevant from an industrial and a clinical standpoint.

A technique with overall good practical performance, Thompson sampling, has been extended to effectively tackle this setting. This contribution is novel for several reasons:

- A proper Bayesian (albeit approximated) treatment of the present data generation process was lacking in the literature, meaning one had to resort to ad hoc sampling methods, like OTFLinTS by Vernade et al. [2020a] which, as mentioned in section 4, performed worse in all the experiments, or to sacrifice having a posterior on delays in order to have a closed form posterior on conversion probabilities, as Wang et al. [2022].

- The present setting is significantly different from the simpler ones in which BTS is commonly applied (in particular, Bernoulli bandits): this has required combining BTS with an EM model, something which, to the best of our knowledge, is novel. In fact, EM effectively performs MLE without the need for a closed form for the likelihood.

- Although the algorithm admits both parametric and non-parametric estimators, we chose to use the latter,

in order to have a model of delays flexible enough to accommodate heavy-tailed distributions. Even if it is a classic technique, the Kaplan-Meier estimator was never applied in the present setting, presumably due to the difficulty of using it in the Bayesian domain. This model can be used in conjunction with commonly available Machine Learning libraries.

- The resulting algorithm was applied to linear contextual bandits with partially observable rewards for the first time.

The proposed approach was compared to a state-of-the-art algorithm on a manifold of families of delay distributions, letting the parameters that characterise these distributions vary. These distributions cover a wide range of scenarios: some are bounded (even deterministic), while others have infinite expectation and even include $+\infty$ among possible realised values. The proposed approach performs significantly better than the competitor in the vast majority of tested environments, and comparably in the remaining minority. Moreover, the competing algorithm requires some tuning of an hyperparameter, whose best value is affected by the distribution of delays (which is, however, unknown to the agent): on the other hand, the proposed approach was tested with the same configuration on all distributions, without any tuning. Ultimately, even if the building blocks of the present algorithm are well known, the proposed method demonstrated superior performance with respect to previous, more "exotic" methods. In this regard, we are close in spirit to the work by Wu and Wager [2022b], which shows that vanilla Thompson sampling shows better performance in dealing with delayed feedback than many techniques that were explicitly developed for that setting.

### 5.2 LIMITATIONS

While reaching significantly lower regret than the state of the art in most studied settings, its execution is admittedly slower, as the EM algorithm requires fitting two MLEs for several iterations before reaching convergence. Depending on the application, this may or may not constitute a problem. Nevertheless, it would be interesting to study an incremental variant of the proposed algorithm, adapting the *Ensemble sampling* technique of Lu and Van Roy [2017].

Moreover, the model was tested assuming that delays are independent of the context/arm. Note, however, that the derivation of the model itself makes no restricting assumption on the dependence of delays on context: it is only when specialising it to the Kaplan-Meier estimator that this choice is made. As seen above, besides delay-context dependence, the model can be easily generalised to tackle non-contextual bandits and contextual non-linear bandits.

The agnostic aspect of the Kaplan-Meier estimator was leveraged to accommodate delay distributions that are very

different among each other. However, this expressiveness could prove detrimental for performance if the experimenter can place strong assumptions on the delay distribution. In these cases, using the given model with a parametric family of distributions could prove more rewarding. Again, due to the general nature of the EM model, this should entail no additional effort.

Finally, this approach makes heavy use of the times between action and observed reward. If these times are not available, this approach would require significant modifications.

## 5.3 FUTURE DIRECTIONS

Besides the extensions mentioned in the previous section, some promising avenues of research emerged, which will be addressed in future work. The first stream regards non-stationarity of the distribution of rewards. Non-stationarity can be retroactively taken into account using a sliding window, beyond which old data points are discarded. However, in the presence of delays, this would effectively induce censoring on delays that exceed the window size, and as seen in simulations censoring can be detrimental for performances, if the window size is too small. This suggests several compelling streams of research. One is adapting, to this partially observable setting, the techniques introduced by Vernade et al. [2020b] and McDonald et al. [2023], that account for the multiple touch points in the marketing funnel. Another is studying whether the approach of Wu and Wager [2022a], which takes into account a non-stationary baseline hazard, can be extended to the present setting. Moreover, as noted also by Vernade et al. [2020b], it would be worth exploring non-stationary bandit techniques besides the sliding window, like an adaptive window size that takes into account how fast the environment changes.

As seen in section 1.2, a number of different approaches exists [Gael Manegueu et al., 2020, Wu and Wager, 2022a, Wang et al., 2022] to treat finite-armed bandits with partially observable delayed rewards. It would thus be of interest to test the approach hereby proposed also in the finite-armed setting comparing it to the other available methods.

Finally, an intriguing setting is that of aggregated, anonymous feedback of Wang et al. [2021]. In a sense, this setting brings the partial observability of conversions to the extreme: an EM approach, using and then eliminating unobserved variables, could prove beneficial also in this harder setting.

## Author Contributions

MG did conceptualisation, literature review, software, formal analysis, data curation, visualisation and original draft preparation; MG and FS gave methodology; FS supervised the study and did review and editing.

## Acknowledgements

This work was partially supported by the MUR under the grant "Dipartimenti di Eccellenza 2023-2027" of the Department of Informatics, Systems and Communication of the University of Milano-Bicocca, Italy.

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

# Bootstrap Your Conversions:
# Thompson Sampling for Partially Observable Delayed Rewards
# (Supplementary Material)

**Marco Gigli**[1]                     **Fabio Stella**[1]

[1]Department of Informatics, Systems and Communication, University of Milano-Bicocca, 20126 Milan, Italy

## A   ADDITIONAL SIMULATION RESULTS

Here we report additional plots depicting the results, in terms of regret, of the simulations that compare our proposed algorithm with the state of the art. In particular, figure 4 refers to simulations in the presence of fixed, deterministic delays; figure 5 refers to the possibility of never being able to observing the feedback, for varying probability $p$; figure 6 refers to delays distributed uniformly on a bounded interval; figure 7 refers to delays distributed according to the Student's $t$ distribution; figure 8 represents a sensitivity test varying the size $n_{\text{prior}}$ of the artificial dataset. Each setting is commented upon in section 4.

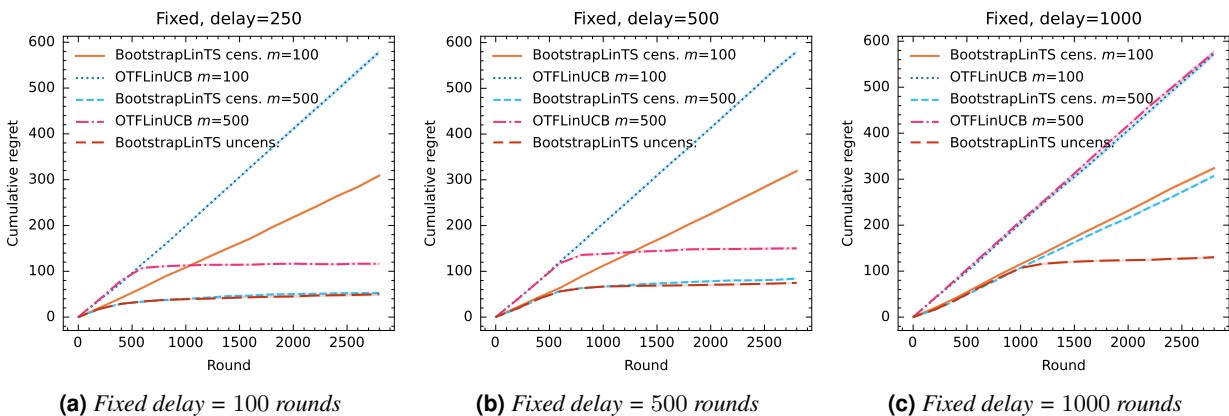

**(a)** *Fixed delay = 100 rounds*          **(b)** *Fixed delay = 500 rounds*          **(c)** *Fixed delay = 1000 rounds*

**Figure 4:** Average cumulative regret suffered by the examined algorithms when delays are fixed.

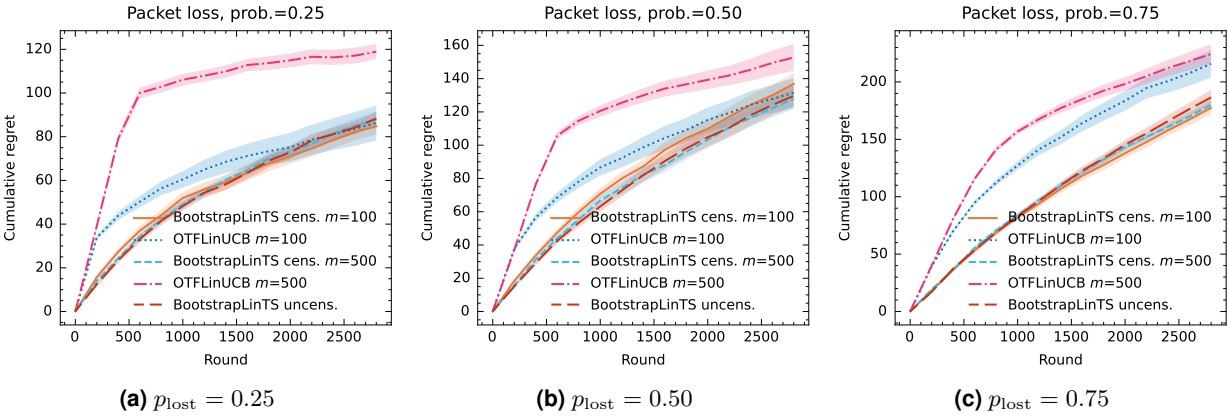

**Figure 5:** Average cumulative regret suffered by the examined algorithms when delays are distributed according to a packet loss distribution, with varying probability $p$ of losing the packet (i.e. of having infinite delay).

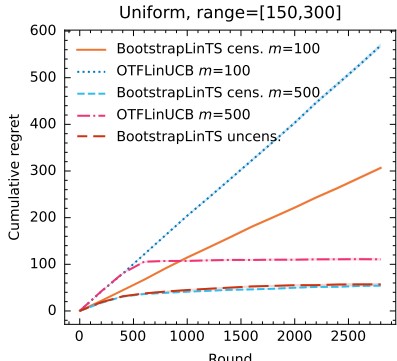

**Figure 6:** Average cumulative regret suffered by the examined algorithms when delays are uniformly distributed.

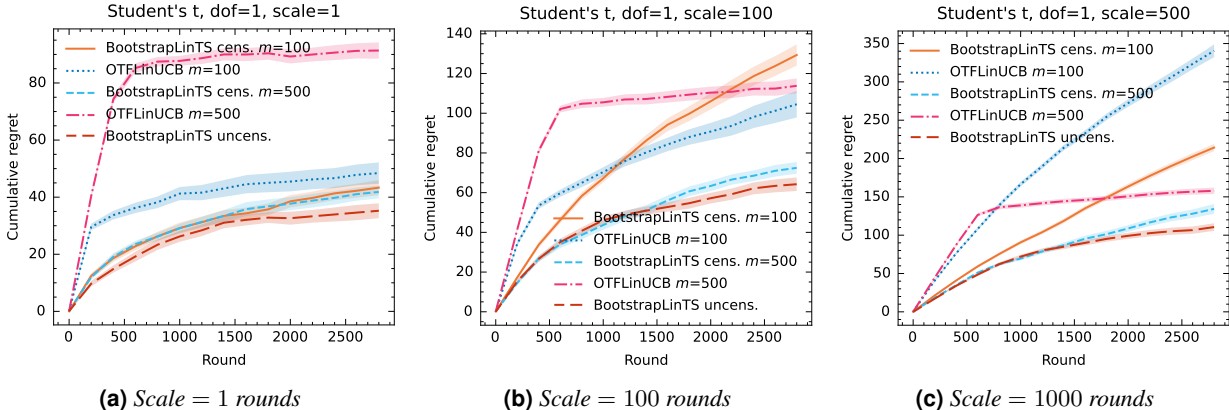

**Figure 7:** Average cumulative regret suffered by the examined algorithms when delays are distributed according to a Student's $t$ distribution, with varying scale and fixed number of degrees of freedom.

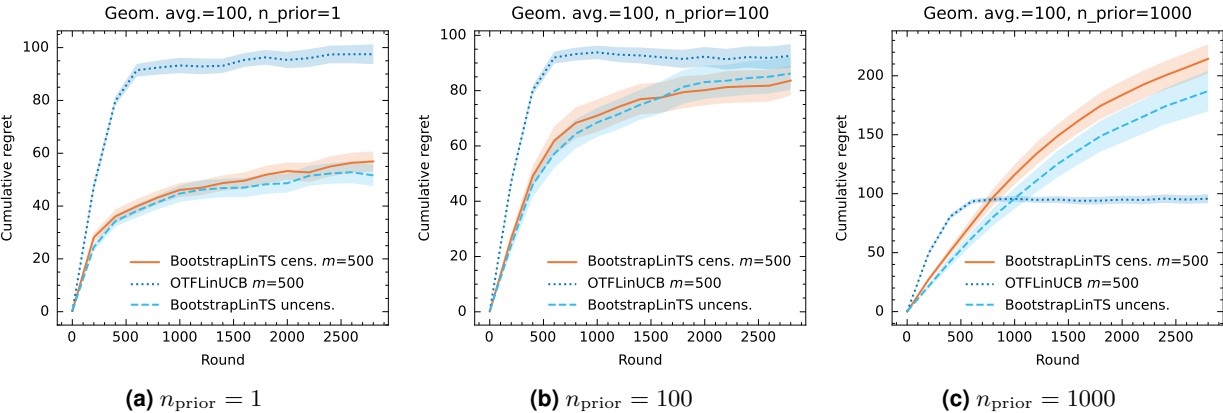

**Figure 8:** Average cumulative regret suffered by the examined algorithms for geometrically distributed delays with average 100, letting $n_{\mathrm{prior}}$ vary.