# OpenReview forum: "Bootstrap Your Conversions: Thompson Sampling for Partially Observable Delayed Rewards"
_auai.org/UAI/2024/Conference — UAI 2024 poster_

### Official Review · Reviewer_1GFK · 2024-03-22

**Q2-1 Originality-Novelty:** 2
**Q2-2 Correctness-Technical Quality:** 3
**Q2-5 Clarity Of Writing:** 3

**Q1 Summary And Contributions:**

This paper presents a Thompson sampling approach when there is delay in the feedback in a contextual bandit setting. Delays and censorship in feedback are common in many situations such as online advertising/shopping.
Through simulations the paper illustrates that the performance of the algorithm is superior to state of the art methods, using multiple test settings.

**Q2-3 Extent To Which Claims Are Supported By Evidence:**

3: Good: the main claims are supported by convincing evidence (in the form of adequate experimental evaluation, proofs, (pseudo-)code, references, assumptions).

**Q2-4 Reproducibility:**

3: Good: key resources (e.g. proofs, code, data) are available and key details (e.g. proofs, experimental setup) are sufficiently well-described for competent researchers to confidently reproduce the main results.

**Q3 Main Strengths:**

1. The paper addresses an important aspect of many bandit use cases i.e. delay / potential censorship in getting feedback.
2. The experimental evaluation is thoroughly done with a range of scenarios. The proposed algorithms consistently shows better performance as compared to benchmarks.

**Q4 Main Weakness:**

1. The technical contribution in the paper is hard to follow. In particular, it seems like the BTS technique has been applied to the current problem and there isn't much novelty in the algorithm design.
2. The paper assumes that the reader is well versed with details of Vernade et al [2020a]. This makes it hard to follow some sections.

**Q5 Detailed Comments To The Authors:**

1. S(T) should be explicitly defined in the paper.
2. It will be helpful if the writing of the paper can be improved. Reducing dependence on the reader being familiar with other papers to follow the main ideas in the paper will be helpful.
3. Making simulation code available will increase the reproducibility of the work.

**Q9 Complying With Reviewing Instructions:**

Yes

---

> ### Author Rebuttal · Authors · 2024-04-07
>
> We thank the reviewer for the effort of reviewing our paper. We hope the following response addresses the concerns expressed, especially regarding novelty.
>
> Q4
>
> 1. We apologise we were not clear enough, and we thank both the present reviewer and the others for giving us the opportunity to produce a clearer final version of the paper. Although one could rightfully say that the contribution of the paper was applying BTS to the current problem, we think this contribution is novel for several reasons. First, the present setting is significantly different from the simpler settings in which BTS is commonly applied (in particular, Bernoulli bandits): this has required testing it with an EM model, something which, to the best of our knowledge, is novel. Moreover, changing perspective, a proper Bayesian (albeit approximated) treatment of the present data generation process was lacking in the literature, meaning one had to resort to ad hoc sampling methods, like OTFLinTS by [Vernade et al 2020a] which, as mentioned in Section 4, performed worse in all the experiments or the method by [Wang et al. 2022] which sacrificed having a posterior on delays in order to have a closed form posterior on conversion probabilities. Finally, although the algorithm admits both parametric and non-parametric estimators, we chose to use the latter: even if it is a classic technique, the Kaplan Meier estimator was never applied in the present setting, presumably due to the difficulty of using it in the Bayesian domain. Ultimately, even if the building blocks of the present algorithm are well known, the proposed method demonstrated superior performance with respect to previous, more “exotic” methods. In this regard, we are close in spirit to [Wu and Wager 2022b], which shows that vanilla Thompson Sampling when dealing with delayed feedback shows better performance than many techniques that were explicitly developed for that setting.
> 2. In the final version of the paper, we plan on adding the needed details to have a more self-contained paper.
>
> Q5
>
> 1. Indeed, we defined the quantity at the end of page 3 (survival function), but forgot to explicitly mention S(T). We thank the reviewer for pointing it out, we will fix it in the final version.
> 2. Please refer to Q4.
> 3. The code was submitted together with the paper and will be made publicly available upon acceptance.

---

### Official Review · Reviewer_mNat · 2024-03-23

**Q2-1 Originality-Novelty:** 3
**Q2-2 Correctness-Technical Quality:** 3
**Q2-5 Clarity Of Writing:** 3

**Q1 Summary And Contributions:**

This experimental paper proposes a new approach for dealing with partially observable delayed rewards in contextual bandits. This approach relies on Bootstrapped Thompson Sampling, a technique in which one applies Thompson sampling, a technique for approximate sampling from the posterior distribution. Although they point out that multiple alternatives exist, they choose to focus on Statistical Bootstrap which gives an estimate on the underlying of the uncertainty parameters by resampling the dataset with replacement many times, and applying the estimator on the resampled data. Then, this method samples from the outcoming distribution of estimates, as if they represented the posterior distribution.
EM is here used as a subroutine of the Bootstrapped Thompson Sampling algorithm. The paper provides an efficient way of performing EM.

To validate this approach, numerous experiments have been conducted, showing the general superiority of the new method with respect to previously proposed methods.

**Q2-3 Extent To Which Claims Are Supported By Evidence:**

3: Good: the main claims are supported by convincing evidence (in the form of adequate experimental evaluation, proofs, (pseudo-)code, references, assumptions).

**Q2-4 Reproducibility:**

4: Excellent: key resources (e.g. proofs, code, data) are available and key details (e.g. proof sketches, experimental setup) are comprehensively described for competent researchers to confidently and easily reproduce the main results.

**Q3 Main Strengths:**

The task at hand is well-motivated and documented. Authors do a good job at relating their work to previous ones, and at documenting past efforts to provide good solutions for dealing with partially observable delayed rewards.
The proposed solution does not make assumptions on the distribution of delays, and hence leaves a lot of liberty. The experiments show a diverse set of possibilities for the way in which both delays and conversions are drawn.
The supplementary material contains the code used for the experiments, a precious asset, that also makes the work reproducible.

**Q4 Main Weakness:**

The paper refers to existing works all along, which is a good point because it provides points of references and explains why this approach has been chosen, and the settings of the experiments. However, this also makes it an arduous read.
Regarding the organization of the paper and its writing, it is difficult as a reader to understand why the technical part of the paper starts with EM. I suggest helping the reader by explaining how the paper is organized and why.
Furthermore, as the key contribution of the paper is to propose an algorithm, I would also suggest outlining it, as simply as possible, but completely.

**Q5 Detailed Comments To The Authors:**

See comments above.

**Q9 Complying With Reviewing Instructions:**

Yes

---

> ### Author Rebuttal · Authors · 2024-04-07
>
> We thank the reviewer for the effort of reviewing our paper. We are glad to see the positive view on our paper.
>
> Q4
>
> We share the reviewer’s opinion that the continued reference to previous works interrupts the flow, making it a somewhat harder read. However, given the tight space constraints, we regrettably had to sacrifice some fluidity to be able to place our work in the right context: upon acceptance, we plan to critically revise the writing and use part of the extra pages to recover some of the lost fluidity, properly introducing each section and the structure of the paper as a whole. In particular, in a previous version of the paper the algorithm was indeed outlined in the main text, but we had to remove the description due to space constraints. In the final version of the paper, it will look as follows:
>
> >We can thus recap the BTS algorithm at round $n$:
> > * The agent receives a dataset of $n$ rows, where each row contains the observable information explained in subsection 2.1;
> > * The agent draws $n_{\mathrm{prior}}$ artificial data points, according to the procedure just described;
> > * The artificial and real data points are merged to form a unique dataset of size $(n + n_{\mathrm{prior}})$;
> > * The agent samples the entire dataset once with replacement;
> > * The model described in section 2 is fit on this sampled dataset, yielding an estimate $\hat{S}(t)$ and $\hat{p}_1(x)$;
> > * The agent plays the action $k$ such that, among $x_1,\dots, x_K$, the context $x_k$ maximises the estimated probability $\hat{p}_1(x)$.

---

### Official Review · Reviewer_Z9fu · 2024-03-23

**Q2-1 Originality-Novelty:** 2
**Q2-2 Correctness-Technical Quality:** 3
**Q2-5 Clarity Of Writing:** 2

**Q1 Summary And Contributions:**

The main contribution of this paper is to propose a way to consider both the uncertainty in the delay and in the conversion in a delayed-conversion framework to improve the exploration-exploitation tradeoff. The method is to apply BTS on the EM algorithm.

**Q2-3 Extent To Which Claims Are Supported By Evidence:**

3: Good: the main claims are supported by convincing evidence (in the form of adequate experimental evaluation, proofs, (pseudo-)code, references, assumptions).

**Q2-4 Reproducibility:**

3: Good: key resources (e.g. proofs, code, data) are available and key details (e.g. proofs, experimental setup) are sufficiently well-described for competent researchers to confidently reproduce the main results.

**Q3 Main Strengths:**

Novelty: In prior works, people considered engineering ways to balance the exploration-exploitation tradeoff in the delayed-conversion setup. This is mainly because the posterior learning of the parameters is not straightforward. The EM algorithm is proposed in Chapelle [2014] as an effective way to find point estimates. BTS proposed a TS-inspired algorithm to include the uncertainty in a point estimate. Thus, the main novelties of this paper are two: 1) in terms of the delayed-conversion setup, the proposed BTS-EM algorithm does not need to engineer an algorithm but can directly consider the model assumption, and 2) the paper claimed that it is the first work to combine BTS and EM.
Correctness, evidence, and reproducibility: The idea of combining BTS and EM makes intuitive sense. With this idea, reproduction should be easy. The authors ran several delayed-conversion examples to show that their algorithm outperformed the previous works.

**Q4 Main Weakness:**

Novelty: In some sense this work is novel as explained in Q3 in terms of the progression in the subfield, but in some sense, this work isn't novel. EM has been proposed before. BTS has been proposed before. Combining BTS and EM seems obvious.
Clarity: This paper is generally very hard to read. While I am familiar with BTS, I am not familiar with the delayed-conversion setup. With the current writing, it was very hard for me to understand the problem setup without looking up prior works. See Q5. A part of it is due to a lack of introduction of notations. A part of it is due to the structure. A part of it is due to the language.

**Q5 Detailed Comments To The Authors:**

Dear authors, I would recommend improving the clarity in the future. Let me explain why I am confused and then provide some suggestions.
1. When I was reading Section 2.1, I was very lost. It started with "Each past interaction with a user (e.g., showing an advert) is associated to a context x that describes the action taken and, optionally, the user itself." I was not even clear what the decision time is, what the actions are, what the reward is, and what we are trying to optimize. I need to guess whether interaction has the same meaning as an action. Why will x (state?) describe the action taken but not A? It might be super helpful to clearly state the problem setup rather than assuming that the readers know. I eventually got it by reading Chapelle [2014].
2. "Moreover, we assume the agent has access to the time δ elapsed since the interaction." What does elapse mean? What do you mean by having access? If you have described clearly what conversion means, it will then be easier to explain what you mean by elapsed time and by having access to it.
3. When you described "A time T , which is the time between interaction and conversion if Y = 1, and elapsed time δ otherwise," since I didn't know what elapsed time was, I had no clue what T was. It will be helpful to explain the conceptual definition of T.
4. In Eq(1), S(T) suddenly showed up. It will be helpful to explain what S(T) is.
5. Below Eq(2), you suddenly mentioned maximizing at step k. I imputed in my head that because it was an EM algorithm, it was iterative and step k means the k-th time to do the maximization. It will be very helpful to clearly explain the E-step, the M-step, and what you meant by step k. Otherwise, the term "step k" just suddenly appeared.
6. I saw that you were trying to explain the EM algorithm in Section 2.3. I am actually quite confused with all the English. For example, I don't know what "row" means. Sometimes English could be hard to understand but equations can be more universal. If you are worried about the length, you can consider putting the equations in the Appendix.
7. You spent quite a lot of time explaining the background of BTS in Section 3. Since I knew BTS, I knew what part was your contribution and what part was proposed in BTS. I think for a general reader, you may consider reorder the entire paper in that you have one section on the prerequisite (which could include the problem setup and BTS). And then after that you have a section about your algorithm. If you think the EM algorithm is part of your contribution, you give it a subsection, and then you give the application of BTS on EM another subsection.
8. Your environment setup appeared in Section 3 (how theta is in [0,1]^d), which should not be part of the algorithm section. I would suggest environment setup has its own subsection, and then the algorithm has its own subsection. We all know the environment and the algorithm do not need to be the same.
9. When I read the parameter "m" in Section 3, I had zero clue what m is. Then, I need to remind myself that you mentioned that in Section 1. I would suggest either reminding people that this m is mentioned in Section 1, or giving people an intuitive explanation of what m is. Since you compare this parameter m in your experiment, it becomes quite important to explain intuitively what m is and why it affects the result.
10. This is not a suggestion. This is a question. How sensitive is your model to n_{prior}? Why do you choose it to be 10 for all simulation testbeds? I think it's directly related to how much you want to regularize (or more precisely, exploration versus exploitation). It might make more sense to tune it according to the environment. What's your thought?

**Q9 Complying With Reviewing Instructions:**

Yes

---

> ### Author Rebuttal · Authors · 2024-04-07
>
> We thank the reviewer for the effort and the detailed feedback, which will greatly enhance readability. We hope that what follows addresses the comments on novelty and clarity.
>
> Q4
>
> Novelty: We agree that, in hindsight, combining BTS and EM may seem obvious. However, we think the fact that previous works had to recur to involved techniques or to sacrifice a significant part of the uncertainty testifies that casting the problem in the Bayesian setting is no trivial effort. This may partially owe to the fact that, despite its simplicity, BTS is less common in the literature than one would expect: we hope the present paper helps show its broad reach. With this goal, we have shown how to make the problem amenable to off-the-shelf ML libraries; using the classic KM estimator was crucial: even we were surprised this EM model was never used with KM. Finally, we think the increased performance versus the SOTA proves the value of the proposed technique.
>
> Clarity: Please refer to Q5.
>
> Q5
>
> 1. We fully agree and reviewer zqRH made a similar remark. We chose a very dry presentation due to space constraints: however, this section would greatly benefit from a rewriting. Besides a few introductory lines, in the final version we will describe the action-feedback mechanism in detail: how it describes the interaction of a website with a user, the available information (actions, contexts), the Boolean reward (pointing to the definition of conversion) and the goal of the optimisation, the delay and the observable quantities. Regarding contexts, we chose the notation x because it is commonplace in the contextual bandit literature. We will use A as an index spanning available actions: x(A) will be the associated context.
> 2. In the final version, we will more clearly state the link between the bandit terminology (reward / feedback) and conversions, defined in the Introduction. To improve clarity, we will write something along the lines of “We assume the agent can keep track of how much time has passed since it has interacted with the user / made the decision”. We will add a footnote which explains why this assumption is at odds with [Vernade et al 2020a] and when it might break.
> 3. We agree that the conceptual reason behind this definition of T is somewhat hidden. It aims at capturing all the information available on the time between a past decision (showing a webpage) and feedback, if any: if feedback was received (the user has converted), the agent records this time; if there is no feedback yet, the agent records the tightest lower bound available (the time between action and current round). In the final version, we will improve the explanation and re-add the piecewise definition of T, removed due to space constraints.
> 4. We are thankful for pointing this out. We defined the survival function at the end of page 3, but forgot to explicitly mention S(T): this will be fixed.
> 5. We agree: we sacrificed some clarity due to space constraints, hence requiring more effort on the reader; in the original version we had the separation E-step / M-step and a proper definition of k. These will be re-introduced in the final version.
> 6. Besides space constraints, in Section 2.3 we did not want to obstacle the reader (especially practitioners) with a wall of equations, which get rapidly cumbersome if all indices are explicit. We understand we obtained the opposite effect, obscuring the methodology. We plan re-adding equations, aiming for a better balance. Moreover, we will substitute “row” with “observation”: we had in mind the rows of a ML dataset, but we agree that “observation” or “data point” are more immediate.
> 7. We introduced BTS to have a self-contained exposition, instead of pointing the reader to [Osband and Van Roy 2015]. However, we agree that distinguishing which part is novel is harder for non-experts. In the final version, we will reorder the material as suggested.
> 8. We thank the reviewer for this remark: we, too, feel that the environment deserves a separate subsection.
> 9. We fully agree on this: in the final section we will point to the original definition of m and intuitively explain its effects.
> 10. The reviewer is right in the interpretation of n_{prior}. We assumed no knowledge of the environment: this guided both the choice of the KM estimator in place of parametric ones and fixing n_{prior} once and for all. The logic is that, if one and the same parameter choice yields good performance across many environments, one can safely take the chosen configuration as part of the algorithm. On the other hand, we agree that a sensitivity test would be interesting: we plan on testing the dependency in the final version.
>
> Q7
>
> We share the view that EM+BTS should not be limited to the present setting. While we could not bring evidence of this broader applicability as it would be outside the scope of our work, we hope this paper can contribute in rekindling the interest for BTS, especially after the suggested improvements.

---

### Official Review · Reviewer_2T2o · 2024-03-23

**Q2-1 Originality-Novelty:** 3
**Q2-2 Correctness-Technical Quality:** 4
**Q2-5 Clarity Of Writing:** 4

**Q10 Ethical Concerns:**

No ethics concerns

**Q1 Summary And Contributions:**

The paper addresses contextual bandits with delayed feedback under partial observability. The data likelihood is modeled accounting for delayed feedback using survival functions to model the delay distribution, combined with modeling the reward distribution given conversion. This is combined with bayesian thomson sampling to accurately model sources of uncertainty without making strong parametric assumptions in the underlying data-generating mechanism. The proposed Boostrapped TS algorithm is evaluated on multiple variants of the contextual bandit setup including analysis with the criteo data to demonstrate utility.

**Q2-3 Extent To Which Claims Are Supported By Evidence:**

3: Good: the main claims are supported by convincing evidence (in the form of adequate experimental evaluation, proofs, (pseudo-)code, references, assumptions).

**Q2-4 Reproducibility:**

2: Fair: key resources (e.g. proofs, code, data) are unavailable but key details (e.g. proof sketches, experimental setup) are sufficiently well-described for an expert to confidently reproduce the main results.

**Q3 Main Strengths:**

1. Contextual bandits with delayed feedback and partial observability is an important problem, so the overall paper is well motivated.
2. Related work is well described and critical gaps are correctly highlighted.
3. Paper is overall well organized and written.
4. Simulations are well conducted and cover a diverse range of possibilities in contextual bandits
5. Empirical results demonstrate utility of the proposed TS variant combined with the proposed MLE that accounts for the data-generating process.

**Q4 Main Weakness:**

1. I am unsure if one real-world evaluation with Criteo data is sufficient evidence of the utility of the proposed method.

**Q5 Detailed Comments To The Authors:**

Authors suggest that their method is general and does not make parametric assumptions. However they do not connect this well with the consequences in MLE estimation or the TS procedures. Specifically, all examples are exponential family type settings based on prior work where posterior distribution can be parametrized in closed form. The paper currently does not describe the implications for other choices of parametrizations, how the identifiability issue is affected by this, and the implication for the proposed TS algorithm.

This is also not well tested in the empirical evaluation. As a result, I am not sure of the overall claim made in the paper. The focus on Kaplan Meier estimator for survival analysis suggests that these are not parametric, but also not contextual? Isn't this unrealistic? There are many advances in survival analysis literature at this point and authors should at least discuss and justify the choice of KM estimator for survival analysis.

I would like to see more real-world evaluation in addition to the synthetic data evaluations.

**Q9 Complying With Reviewing Instructions:**

Yes

---

> ### Author Rebuttal · Authors · 2024-04-07
>
> We thank the reviewer for the effort of reviewing our paper. We hope the following response addresses the concerns regarding generality and impact.
>
> Q4
>
> We had the same doubt, and we would have included other real-world evaluations if we knew of other publicly available datasets. Due to this, we have included simulations on a wide variety of delay distributions: since we found our method performing well on these, we are confident that it would be able to handle new real-world dataset. If the reviewer has any suggestion on other real-world datasets, we are happy to include them in the final version.
>
> Note that, for instance, we cannot use the same data employed by [Wu and Wager 2022a], as they test their method on a semi-synthetic experiment: they simulate data using the same model they are testing (i.e. Cox), and only baseline hazards are taken from real-world data. As touched upon in the Related Work section, their setting is that of time-dependent hazard (outside the scope of our paper), but with no dependence on time since exposure, which in our setting translates to the already present geometric distribution.
>
> Q5
>
> Exponential family
>
> We remark that, while for all exponential family distributions the posterior can be expressed in closed form, this cannot be done in our case (even though also the Bernoulli distribution belongs to the exponential family) due to the tight coupling between delays and conversions. In particular, the agent does not observe a delay variable and, separately, a Bernoulli variable (the latent variables C and D in the notation of the paper), but rather censored variables Y and T. Indeed, in a previous version of the paper, we mentioned the following (we had to remove this paragraph due to the tight space constraints):
>
> In our setting (the same considered by [Vernade et al. 2020a]), the environment is described by two probability distributions (for delays and conversions respectively). Thus, the posterior would be a distribution of distributions. Even if the prior over delay distributions were independent from the prior over conversion distributions, independence would be lost in the posterior. For instance, the same data could be interpreted either as having already seen most conversions (belief peak on relatively small delays and low conversion probability) or as still missing most conversions (belief peak on high delays and conversion probability): the posterior would not factorise. We thus share the reservations expressed by [Vernade et al. 2020a] regarding the possibility of efficiently computing such posterior distribution.
>
> We also remark that, strictly speaking, fixed delays, uniformly distributed delays and packet-loss delays are not of the exponential family type, although we concede that the first two distributions are very well-behaved (so much so that the first is not even a distribution) and the third is closely related to Bernoulli distributions (which form an exponential family). With this in mind, we plan to extend our tests to the Student’s t distribution: due to the time limit during the rebuttal period, we leave it for the final version.
>
> Finally, we note that the choice of the distributions covers those of previous works (Lancewicki et al. 2021 and Wu and Wager 2022b) because we felt this manifold of distributions is becoming a standard benchmark in this field. That said, we are happy to extend this set in the final version.
>
> Real-world evaluation: Please see Q4
>
> KM estimator
>
> We will add further comments on the choice of the KM estimator in the final version. We chose it because (although being a rather old technique) it is still the de-facto workhorse for non-parametric Maximum Likelihood estimation of censored time data, when dealing with univariate models (i.e. not contextual). The choice fell on a non-parametric estimator since a strength of the proposed method is being distribution-agnostic, as it works well on a wide range of distributions, without the need to know them beforehand. The use of a univariate model, dealing with bandits with partially observable delayed feedback, is justified either when the set of actions is that of a vanilla bandit (i.e., no contexts at all) and each action has its own delay distribution, or when (as in our case) there is one delay distribution, independent of the action. As touched upon in Section 3, this choice was justified by [Vernade et al 2020a] studying the Criteo dataset. Besides this, we wanted to place ourselves in the same setting as that work, in order to have a benchmark technique. However, as remarked in Section 2.3, the model can be extended straightforwardly to context-dependent delay distributions: being outside the scope of the present paper, we leave this extension to future work. Finally, besides its theoretical underpinnings, the KM estimator has the non-trivial advantage of being implemented in off-the-shelf Machine Learning libraries, favouring reproducibility and adoption by practitioners.

---

### Official Review · Reviewer_zqRH · 2024-03-24

**Q2-1 Originality-Novelty:** 3
**Q2-2 Correctness-Technical Quality:** 3
**Q2-5 Clarity Of Writing:** 3

**Q1 Summary And Contributions:**

The paper proposes Boostrap TS that combines with EM algorithm to deal with bandit with partially observed and delayed feedback.

**Q2-3 Extent To Which Claims Are Supported By Evidence:**

3: Good: the main claims are supported by convincing evidence (in the form of adequate experimental evaluation, proofs, (pseudo-)code, references, assumptions).

**Q2-4 Reproducibility:**

3: Good: key resources (e.g. proofs, code, data) are available and key details (e.g. proofs, experimental setup) are sufficiently well-described for competent researchers to confidently reproduce the main results.

**Q3 Main Strengths:**

Originality: It is true that such problem has been investigated before and none of the pieces are completely novel, but I found the idea of combining these pieces to better deal with delay is very interesting and to me a pleasure to read.

Technical quality: The derivation details of the EM and the whole point to point comparison with existing work is very clear. The experiments are also pretty comprehensive.

Clearity of writing: I found it clear to follow the paper and appreciate the idea.

**Q4 Main Weakness:**

Writing: I think it is better to have a well defined problem setup in an algorithm box rather than describing it. For instance, do we observe contexts for all arms at each round? I am not very clear.

**Q5 Detailed Comments To The Authors:**

1. In the algorithm, we sample artificial history at every step. Is this really necessary? Can we just sample at the beginning?
2. As said above, do we observe contexts for all arms like in Vernade 2020a? I thought we just observe a context vector, select an arm and then observe (delayed) feedback. Maybe it is better to articulate the setup more clearly at the beginning.
3. Does using more bootstrap samples at each step help?
4. For the experiments, can you increase the number of replications? Some of the error bars seem huge to decrease readability of the plots.

**Q9 Complying With Reviewing Instructions:**

Yes

---

> ### Author Rebuttal · Authors · 2024-04-07
>
> We thank the reviewer for the effort of reviewing our paper. We are glad to see the positive view on our paper.
>
> Q4
>
> We are thankful for pointing out a lack of clarity in Section 2.1: we agree and, indeed, this opinion is shared also by reviewer Z9fu. Besides space constraints, we think this is mainly due to the perspective we chose, of describing directly the dataset available at a given round (hence speaking of “each past interaction with a user”). We feel we would be clearer describing in greater detail the agent-user interaction: we will use the extra space in the final version for this and for introducing the section more gently. We explain the details here and will expand on these in the final version. We use the website optimisation setting for concreteness, but the concepts can be easily mapped to the other settings mentioned in the Introduction.
>
>
> Whenever a new user comes to the website, a new round of the optimisation begins: the agent must decide among $K$ page variants, which constitute the available set of actions for that round. We assume the agent observes a context $x(A) \in \mathbb{R}^d$ for every action $A=1, \dots, K$, prior to choosing the action.
>
>
> This assumption accommodates three scenarios: i) users are described by features (interests, location, demographics, device and so on) and web pages are not; ii) web pages are described by features regarding their content, while users are not; iii) both users and web pages are described by features. In case ii), the vector of features of web page $A$ is directly its context $x(A)$. In case iii), the context $x(A)$, which represents the interaction of a web page with a user, can be obtained by the outer product of the user feature vector and the web page feature vector. Case i) can be recovered as a limiting case of the third one, associating to each web page the corresponding canonical basis vector $e_A$ of $\mathbb{R}^d$. We agree we did not discuss contexts to a great length, as we assumed the interested reader to have a high degree of familiarity with contextual bandits.
>
>
> We assume that, right after the agent shows the page to the user, two latent variables are generated: a Boolean variable $C$, which indicates whether the user does convert or not, irrespective of when; a latent time variable $D$, the delay between being shown the page and conversion (undefined or infinite if $C=0$).
>
>
> After a time $\delta$ has passed since the agent-user interaction, the conversion may or may not have happened already: we thus introduce a Boolean variable $Y$, which indicates if a conversion has been observed by the agent. We also assume that the agent can keep track of the time $\delta$ and, if the conversion has indeed happened, of the time $T$ between serving the web page and the conversion. If the conversion did not happen (either because it will happen in the future or it will not happen at all), the time $T$ is simply $\delta$, the lower bound to the interaction-conversion time.
>
>
> Q5
>
> 1. This is a very good question. We think there are good reasons for sampling a new artificial history at every round. Sampling just once at the beginning means adding some data points to the dataset (although from the “prior” distribution rather than from the environment): in a way, this means just shifting the clock a few rounds ahead, but other than that it is very similar to not sampling an artificial history at all. In particular, this could be damaging in two opposite cases: when the sampled artificial history is “too much well-behaved” the agent underestimates uncertainty; when the artificial history is “pathological”, the agent is stuck with an artificial dataset which does not represent the environment well.
> 2. Please see answer to Q4.
> 3. We could not decide between two interpretations of “more bootstrap samples”, so we considered both: 1. Sample with replacement e.g. 2*n data points from a dataset with n observations; 2. Sample with replacement n data points, but doing so many times. Case 1 would not comply to the standard Statistical Bootstrap, and it would not be beneficial intuitively since it would artificially reduce the uncertainty: the goal is estimating the degree of variability of the estimated quantities, given a dataset which has as many observations as the present one. As for case 2, while usually sampling many times is required for estimating said variability, here we need just one resampling, in accordance with TS: we need one sample from the posterior-like distribution, to be used as a representation of the environment for the present round. Sampling many times could be useful for diagnostic reasons, to track how the uncertainty on the estimated parameters evolves over time.
> 4. We thank the reviewer for this remark. We have started increasing the number of replications: due to the time limit of the rebuttal period (and the slow dependence of uncertainty on number of replications) we leave the results to the final version of the paper.

---

### Meta-Review · Area_Chair_A1Ew · 2024-04-11

This paper proposes an approach for contextual bandit problems with partially observable, delayed feedback. Reviewers largely agreed that the work tackles an important problem in a novel way. There was less agreement on the quality of presentation, with some praising it but one complaining that the paper was hard to read. I was convinced by the authors' reply to this complaint in the rebuttal.